# NMR-Based Characterization of Citrus Tacle Juice and Low-Level NMR and UV—Vis Data Fusion for Monitoring Its Fractions from Membrane-Based Operations

**DOI:** 10.3390/antiox12010002

**Published:** 2022-12-20

**Authors:** Martina Gaglianò, Giuseppina De Luca, Carmela Conidi, Alfredo Cassano

**Affiliations:** 1Department of Chemistry & Chemical Technologies, University of Calabria, Via P. Bucci, 87036 Rende, Italy; 2Institute on Membrane Technology, ITM-CNR, 87036 Rende, Italy

**Keywords:** Tacle juice, NMR, UV—Visible, ultrafiltration/diafiltration, nanofiltration

## Abstract

Tacle is a citrus variety which recently gained further interest due to its antioxidant and biological properties. This study suggests using Nuclear Magnetic Resonance (NMR) imaging to characterize Tacle juice’s metabolic composition as it is intimately linked to its quality. First, polar and apolar solvent systems were used to identify a significant fraction of the Tacle metabolome. Furthermore, the antioxidant capacity and the total content of flavonoids, polyphenols and β-carotene in the juice were investigated with UV—Visible spectroscopy. Tacle juice was clarified and fractionated by ultrafiltration (UF) and nanofiltration (NF) membranes in order to recover and purify its bioactive principles. Finally, the second part of this work sheds light on the spectrophotometric assays and ^1^H-NMR spectra of fractions coming from membrane operations coupled with a multivariate data analysis technique, PCA, to explore the impact of UF and NF processes on the metabolic profile of the juice.

## 1. Introduction

Tacle is a new triploid citrus hybrid developed in Sicily using traditional, strictly non-GMO techniques. Both its name and flavour recall the two parents’ cultivars: the Tarocco orange (*C. sinensis* L. Osbeck) and the Monreal Clementine (*C. clementina* Hort. ex Tan.) [1]. This innovative fruit was mainly studied for its high antioxidant activity [2], which confers the citrus extracts’ health-promoting effects of lowering risks for chronic heart and vascular diseases and treating metabolic disorders such as obesity and diabetes [3,4]. The outbreak of the COVID-19 pandemic has increased the already-high consumer demand for immune-boosting, natural and organic products [5]. The high nutraceutical profile of Tacle responds perfectly to this need. Still, the main bottleneck may lie in the juice’s production process, which must allow for the obtaining of processed food very close to the raw material. Hence, apart from the sensory attributes, one should minimize the loss of colour, texture and significantly bioactive compounds. Traditionally, fruit processing techniques include clarification with filter aids and fining agents, as well as thermal processes such as pasteurization, sterilization and concentration aimed at improving the shelf life of the juice by preventing enzymes, spoilage and pathogenic microorganisms. Unfortunately, these processes lead to a dramatic change in the phenolic and carotenoid compounds, vitamins, taste and color of juices, as well as to the formation of undesirable substances such as furfural, 5-hydroxymethylfurfural, furan and acrylamide [6]. Consequently, the fruit juice’s quality is compromised in terms of its nutritional, functional, physicochemical and sensorial properties. Pressure-driven membrane operations have emerged in the past as non-thermal technological alternatives to traditional methods given their ability to minimize the degradation of functional molecules and promote the development of high-quality products with considerable shelf stability [7,8,9]. Among them, microfiltration (MF) and ultrafiltration (UF) provide cost-effective alternatives to traditional fining and clarification methodologies for separating the raw juice into a fibrous concentrated pulp (retentate) and a clarified fraction free of spoilage microorganisms (permeate), which are suspended solids and colloids such as proteins [10]. The combination of these processes with diafiltration (DF) increases the purity of the obtained fractions. For instance, carotenoid extracts from solid by-products of cashew apple (*Anacardium occidentale* L.) juice processing, as well as from orange juice, have been obtained through a combination of crossflow MF and DF [11,12]. Clarified juices can be further fractionated or concentrated by using nanofiltration (NF) and reverse osmosis (RO) operations [13,14]. The environmental friendliness, economics and ease of use are further advantages encouraging the application of these technologies [15]. The evaluation of membrane processes in retaining metabolites which exhibit high nutritional value, shape organoleptic characteristics and thus formulate the fundamental quality properties of the product can be detected by means of several analytical methodologies [16,17,18,19]. NMR spectroscopy is emerging as a leading technique in metabolomic studies, as it is a fast, robust and reliable technique which allows the detection of a wide range of structurally diverse metabolites simultaneously. Because the richness of information often results in high spectral complexity, it calls for using multivariate analysis to study large numbers of spectra and extract meaningful information. Among the multivariate statistical analysis techniques, principal component analysis (PCA) is widely used and is recognized as one of the primary unsupervised compression techniques for exploratory data analysis. Recently, NMR-based metabolomics coupled with chemometric analysis has been applied to obtain metabolic profiles of various kinds of food, including honey, olive oil, apple and pomegranate juice, thus establishing adulteration, variety, geographical origin, quality and authenticity [20,21,22,23]. PCA has also been applied very successfully to study the industrial processing of citrus fruit juices to evaluate for reliable process control, to assess the quality of orange juice and to identify potentially mislabeled samples [24]. Furthermore, even though it is still a scientific challenge, NMR measurements can be combined with metabolomic and statistical data from other techniques. For example, Tristán et al. [25] probed the feasibility of using FTIR and NMR data to achieve, through the models developed, a fast and accessible tool for evaluating the ripe state of the melon fruits. Additionally, fusion of ^1^H-NMR and chromatographic techniques (gas and liquid chromatography) data coupled with mass spectrometry was applied to provide more an accurate knowledge about the classification of golden rums [26]. The combined use of multi-technique data in chemometric analysis produced the best results compared to the individual techniques in classifying and distinguishing samples [27].

In this work, NMR has been applied to the analysis of Tacle juice with multiple aims. A preliminary chemical characterization of the juice was performed by using deuterium oxide (D_2_O) and deuterated chloroform (CDCl_3_) as solvents in order to capture both polar and non-polar metabolites from the original sample, thus improving the coverage of its metabolome. The juice was treated by UF to obtain a clarified fraction free of suspended solids and with a phenolic composition similar to that of the raw juice. The combination with DF was also studied in order to improve the purity of the retentate fraction. A final NF step was performed in order to measure the fraction of concentrated bioactive compounds from the clarified juice. NMR experiments and colourimetric assays were carried out on UF/DF and NF fractions to investigate and monitor how membrane processes affected the metabolic profile of the juice. Antioxidant activity and total content of flavonoids, polyphenols and β-carotene were the variables determined by means of UV—Visible spectroscopy, and were evaluated in combination with NMR measurements for the principal statistical analysis (PCA).

## 2. Materials and Methods

### 2.1. Tacle Juice and Juice Processing

Tacle juice was supplied by Società Agricola Terzeria S.r.l located in Francavilla Marittima (CS), Calabria. The juice was ultrafiltered by using a laboratory plant equipped with a hollow fiber membrane module supplied by Microdyn-Nadir (Wiesbaden, Germany). It featured polyethersulfone hollow fiber membranes with an inner diameter of 0.8 mm and a molecular cut-off weight (MWCO) of 500 kDa. The UF process was performed in selected operating conditions according to the batch concentration configuration in order to clarify the juice to a volume reduction factor (VRF, defined as the ratio between the initial feed volume and the volume of the resulting retentate) of 4.08. At the end of the UF process, two different streams were produced from the original juice (feed UF, abbreviated as FUF): a clarified juice (indicated as PUF) and a retentate juice (indicated as RUF). Then, the retentate was diafiltered in a continuous mode by adding water to the feed tank at the same rate as the permeate flux, so as to keep the feed volume constant during the process. Two fractions were collected at the end of the diafiltration process: a retentate stream (named as RDF) and a permeate stream (named as PDF). Both permeates resulting from ultrafiltration and diafiltration operations were then merged in the same ratio. The mixed product was then subjected to a nanofiltration process by using a laboratory plant equipped with a stainless-steel housing able to accommodate a spiral-wound membrane module with an effective membrane area of 0.38 m^2^. NF experiments were performed in selected operating conditions by using two different membrane modules, both in thin-film composites supplied by Microdyn-Nadir: a membrane module with a MWCO of 200–300 Da (NF1) and a membrane module with a MWCO of 300–500 Da (NF2). The NF processes were performed up to a VRF of 3.5. A schematic layout of the investigated process is depicted in Figure 1. UF/DF and NF processes were repeated five times on five batches of Tacle juice. Each batch, previously frozen at a temperature of −18 °C, was thawed after approximately two weeks to be subjected to membrane processes, and the resulting fractions were then frozen again to be re-thawed for the analysis of chemical characteristics. All samples collected from juice processing, for a total of 50 samples, were spectroscopically and spectrophotometrically analyzed and statistically processed by PCA.

### 2.2. NMR Sample Preparation

The chemical composition of polar compounds in Tacle juice was evaluated on a sample prepared as follows: Tacle juice (FUF) was thawed at room temperature, diluted at a 1:1 ratio with distilled water and centrifuged (6000 rpm × 5 min) to separate the supernatant from the pulp. In addition, the pH of the supernatant was measured since it is crucial for this parameter to be known before the signal’s attribution phase to the various metabolites, whose resonance frequency is strongly dependent on the pH. In this specific case, the pH value of the sample was measured as 3.41. Subsequently, a volume of 500 μL of supernatant was transferred into a 5 mm NMR tube, to which 100 μL of D_2_O and 20 μL of a 100 mM solution of TMSP-d_4_ (sodium deuterated 3-trimethylsilylpropionate) and 20 mM of NaN_3_ in D_2_O were added. Deuterated water was used for locking the signal, 3-(trimethylsilyl)propionic-2,2,3,3-d4 acid sodium salt (TMSP-d_4_) was used for referencing chemical shift, and sodium azide (NaN_3_) was used to prevent the onset of bacteria capable of destroying juice metabolites during the recording of uni- and bi-dimensional multinuclear NMR experiments. The preparation of the NMR samples used for multivariate statistical analysis did not require uae of the internal standard nor the sodium azide, given the much shorter times for recording a single proton spectrum. NMR sample preparation for RUF and RDF were exactly the same as for FUF. These samples were diluted and centrifuged, and then 500 μL of supernatant was added to 100 μL of D_2_O in an NMR tube. PUF and PDF samples were diluted as well but they did not need to be centrifuged, as they lack suspended solids which might interfere with the homogeneity of the magnetic field. Instead, all NF samples were neither diluted nor centrifuged, but weredirectly transferred in the NMR tube with D_2_O, always at a 5:1 ratio. Before transferring the supernatants or samples directly into the NMR tube, the pH value was evaluated for each sample type. Utilizing NaOH or HCl solutions, the pH value of UF samples were adjusted to be the same, just as for NF samples. Equivalent pH values imply the same position in the chemical shift of the signals relating to the metabolites present, which is essential for the steps before and after PCA. A synthetic scheme of the NMR sample preparation in aqueous phase is shown in Figure 2.

On the other hand, to characterize the apolar fraction of Tacle juice metabolome, a simple, efficient and low-solvent-consuming, ultrasound-assisted extraction (UAE) method was investigated utilizing the deuterated solvent CDCl_3_. Then, 1 mL of Tacle juice (FUF) was combined with 1 mL of CDCl_3_ containing 0.01% TMS, and placed in an Ultrasonic Bath (UAE; Hielcher UP 100 Hz, 100 W pulse, 30 kHz frequency) for 15 min. After the extraction, two separate phases were observed: the aqueous phase above, which was removed, and the organic phase below, which was transferred into a 5 mm NMR glass tube. A synthetic scheme of the NMR sample preparation in organic phase is shown in Figure 3.

### 2.3. NMR Data Acquisition and Processing

All spectra were acquired on a Bruker Avance 500 spectrometer operating at 500.13 MHz, 298 K and a magnetic field of 11.7 tesla. To suppress the residual water signal through selective irradiation at the water frequency during the mixing and recycle delays, ^1^H NMR spectra of polar extracts were acquired using a NOESY pre-saturation pulse sequence (Bruker sequence denoted as *noesypr1d*). For each experiment, 512 FIDs were acquired using a spectral width of 11.25 ppm and a relaxation delay of 3 s. For lipophilic extracts, ^1^H NMR spectra were obtained using the Bruker pulse sequence *zg.* The acquisition conditions for CDCl_3_ extracts were as follows: number of scans (NS) = 512; spectral width (SW) = 11.25 ppm; size of FID (TD) = 65,536; relaxation delay (D1) = 3 s. All ^1^H-NMR spectra were phased and then baseline-corrected using the software TopSpin 3.6.2. (Bruker Corporation, Billerica, MA, USA). The obtained spectra were calibrated according to the internal standard. Additionally, the ^1^H^1^H correlation spectroscopy (COSY) and ^1^H-^13^C heteronuclear multiple-quantum coherence (HMQC) spectra were recorded in the aqueous and organic phases to verify metabolites’ chemical shift assignments using the Bruker sequences *cosygpprqf* and *hmqcgpqf*. A sine filter and a qsine filter were applied on both dimensions, F1 and F2, respectively, on the COSY and HMQC experiments before they were Fourier-transformed.

### 2.4. UV—Visible Analysis of Total Polyphenols, Flavonoids, In Vitro Total Antioxidant Activity and β-Carotene

Total polyphenols were measured colourimetrically via the Folin—Ciocalteau method, as reported elsewhere [28]. Gallic acid was used as a calibration standard, and results were expressed as mg gallic acid equivalent (GAE) per liter of sample (mg GAE/L). The total flavonoid content was determined according to the Davis method [29]. Quantification was done on the basis of the standard curve of naringin (r^2^ = 0.990), with the results expressed as mg naringin equivalents (NE)/L. The total antioxidant activity (TAA) was assessed via the 2,2-azino-bis (ethylbenzothiazoline-6-sulfonic acid) (ABTS) assay by monitoring the reduction of the radical cation as the percentage inhibition of absorbance at 734 nm [30]. Results of TAA were expressed in terms of mM of 6-hydroxy-2,5,7,8-tetramethylchroman-2-carboxylic acid (Trolox) equivalent. The concentration of β-carotene was determined using the spectrophotometric method reported by Lime et al. [31] and results were expressed as µg β-carotene/mL. Total content of β-carotene was quantified only in UF/DF samples because as a large and lipophilic compound, we expected to find it mostly in the starting juice and in the UF retentate fractions. Since the subsequent steps of the membrane processes (NF) concerned the use and treatment of the clarified juice, focus was placed on monitoring low molecular weight compounds, which is why β-carotene was not quantified for NF samples. Total phenolics and total flavonoids, having lower molecular weights and TAA, were instead determined for all UF/DF and NF samples. Each analysis on each sample was replicated three times, and the mean values obtained were inserted into the matrix containing the NMR data and considered as variables for the PCA statistical analysis.

### 2.5. Chemometric Analysis and Procedure

NMR aqueous phase spectra recorded on membrane samples were aligned, divided into uniform spectral bins (so-called buckets) of 0.05 ppm-width and the signal area was integrated for each bucket. The water region from δ 4.2–5.1 was excluded from the spectra along with the region from δ 5.5–5.9 which does not contain any significant signal. For the residual regions (δ 0.6–8.4 for UF/DF samples and δ 0.4–9.1 for NF samples), binning was performed. The buckets were normalized to the whole spectral area and exported in an Excel file. A dataset of 25 samples and 127 NMR variables had been obtained for UF/DF processes, while 25 samples and 144 NMR variables were considered for NF processes. On the 50 samples of UF/DF and NF processes, the content of total phenolics, flavonoids and antioxidant activity were quantified. For the 25 samples of UF/DF the total β-carotene was also determined. Adding these variables obtained with the UV—Visible technique, the data matrices used for the chemometric analysis were formed as follows: 25 samples and 131 variables (127 NMR + 4 UV—Vis variables) for UF/DF, 25 samples and 147 variables (144 NMR + 3 UV—Vis variables) for NF. In this way, a large block of variables (NMR variables) dominated over a smaller block of variables (UV—Visible variable) for purely numeric reasons. To address this problem and give each block the same importance, a block-scaling procedure was employed in the pre-processing phase of multivariate data [32]. This corresponds to down-weighting blocks of variables in relation to a selected basis-scaling procedure. The basis-scaling method generally entails scaling to unit variance, mainly when variables differ noticeably in nature and numerical range [33]. In this work, block-scaling treatment was carried out previously and then PCA was performed on the fused matrices, which combined NMR and UV—Vis data. Multivariate statistical analysis was performed with the Chemometric Agile Tool (CAT), an R-based chemometric software developed by the Chemistry Group of the Italian Chemical Society [34].

## 3. Results and Discussion

### 3.1. Tacle Juice Metabolic Profiling by NMR

The characterization of the metabolic profile of Tacle juice through NMR spectroscopy was performed in both aqueous and lipophilic citrus extracts to obtain as much compositional information as possible about major compounds in the tested samples. Figure 4a–c shows three ^1^H-NMR spectral expansions of an aqueous extract of Tacle evidencing the assigned metabolites. Most of them belong to sugars, amino acids and organic acids. As mentioned before, all assignments were based on the analysis of 1D and 2D NMR experiments (reported in the Appendix A) and on the use of HMDB database and literature data [35,36]. Additionally, Figure 4d provides a typical ^1^H-NMR spectrum of an organic extract where the main families of metabolites identified were fatty acids and triacylglycerols (TAGs). A numeric nomenclature is used to identify each signal, following a descending field order. The assignment of the different signals was based on previous reports on the ^1^H-NMR analysis of fats [37,38,39,40] and is summarized in Table 1.

From a purely qualitative point of view, the metabolic profile of Tacle juice appears to be almost identical to those of the parent cultivars [20]. In the aqueous phase, a total of 11 amino acids were identified together with 6 organic acids, citric acid being the most major among them. The most crowded part of the spectrum (3.0–4.3 ppm) contains the carbinolic protons of the monosaccharide residues, particularly α- and β-glucose, α- and β-fructose, and sucrose. The remaining down-field part the proton spectrum (6–8 ppm) is diagnostic of aromatic signals and it may have some importance for glycosides with an aromatic aglycon. Such is the case for phlorin (3,5-dihydroxyphenyl-β-D-glucopyranoside), a long-discussed molecule for the authentication of citrus juices as a possible marker for fraudulent processing techniques [41], whose aromatic protons resonances fall at 6.15 ppm (triplet generated by the para-positioned proton) and at 6.22 ppm (doublet generated by the two magnetically identical, ortho-positioned aromatic protons). The extraction of the lipid fraction of the Tacle juice showed the presence of some very low-intensity signals related to fatty acids, agreeing well with the fact that it is a fruit with very few calories. Fatty acids can be distinguished in the ^1^H-NMR spectrum only at a class level (saturated, monounsaturated, diunsaturated and polyunsaturated), and it is impossible to distinguish between signals of individual fatty chains within the same class. For instance, ^1^H resonances of long-chain fatty acids such as palmitic and stearic acids are completely overlapped. For this reason, many assignments reported in the literature, such as in this work, are referred to as molecular fragments (methyl, methylenic, allylic methylenic groups, double bonds, etc.) rather than individual compounds [42]. Once the original juice (FUF) had been characterized, NMR spectra in the aqueous phase were recorded on the fractions obtained from membrane operations it had undergone. Figure 5 shows the NMR spectral profiles related to ultrafiltration samples (RUF and PUF) and diafiltration samples (RDF and PDF). Visually, these fractions appear to have the same metabolomic profile.

Similarly, the high-resolution NMR spectra of the aqueous fraction were recorded for the samples coming from the nanofiltration processes, reported in Figure 6.

In this case, NMR profiles present differences in some signals and their relative intensities. These differences are especially noticeable in the enlarged region from δ 2.55–2.85 of Figure 6, where the citric, aspartic and malic acid signals fall. In the NF feed, citric acid is present at a higher concentration than the other two acids, as NMR is an intrinsically quantitative technique [43], and with an equal number of nuclei, the area of the two generated doublets of citric acid in the ^1^H-NMR spectrum is higher with respect to other acid signals. On the other hand, in the NF permeates, the same acids are present, but in different proportions: these fractions have a higher aspartic acid content, thus indicating that citric acid is retained much more by NF membranes with than by aspartic acid. However, in NMR spectra of NF retentates, no relevant signals corresponding to citric acid (in the region δ 2.55–2.85) can be detected. Its signals may fall into the NMR spectra background noise. Furthermore, it should be pointed out that citric acid is a Lewis acid consisting of functional hydroxyl groups, which can form complexes with other compounds and hydrogen bonds with the membrane polymer; thus, it could be partially absorbed on the membrane surface. Moreover, with the initially higher concentration of citric acid in the feed solution, its molar mass slightly higher than that of aspartic acid as well as the different pK_a_ values of these organic acid compounds may all contribute to determine a different behavior in terms of NF membrane rejection.

### 3.2. Principal Component Analysis Applied to NMR and UV—Visible Data of UF/DF and NF Samples

To better understand variable correlations, membrane sample similarities/differences, and thus the influence of integrated-membrane processes on the metabolic juice profile, NMR data (block 1) were combined with UV—visible data (block 2) in the principal component analysis (PCA). Combining data from two or more techniques is advantageous if they offer complementary information that, when added together, allows us to interpret the process comprehensively and better distinguish between the samples. For example, if we consider only the NMR variables, there is a cluster separation of UF/DF samples, but not when the UV—Vis variables are added. Figure 7 shows the PCA score plot obtained from autoscaling and processing the data matrix containing NMR data only.

The two principal components in this score plot explain 95.2% of the total variance. We can observe that feed samples differ from the retentate samples (in turn, the retentates from ultrafiltration and diafiltration are separated into two different regions of the diagram). In contrast, the permeate samples separate from the feed and retentate samples but fall into the same region, indicating their similar composition in metabolites identified through NMR. Adding UV—Vis data to the NMR data (Figure 8), UF/DF samples were separated into five clusters with no notable outliers; UF permeates and DF permeates are now distinguished very well. Figure 8 shows the PCA score plot of fused NMR and UV—Vis data matrix pretreated by block-scaling. PC1 described 59.9% variation of data, while PC2 accounted for 33.5% of the total subset variance. Therefore, more than 90% of variation can be described only by the first two PCs, and this is why only PC1 and PC2 were selected for further analysis. The evaluation of the PCA score plot indicated that the clustering between FUF, RUF and PUF samples was along the PC1 variable. These samples all had approximately the same negative value of PC2, but FUF samples had PC1 = 0 and RUF samples were characterized by PC1 > 0, while PUF samples hold PC1 < 0. Thus, the PC1 component allowed for distinguished samples coming from the ultrafiltration. On the other hand, the PC2 component permitted us to discriminate ultrafiltration samples from diafiltration samples. In fact, while the first ones were all at a negative value of PC2, the diafiltration samples (RDF and PDF) had a positive value of PC2 and were separated among them on the PC1; RDF samples were in quadrant 1 (upper right), while the PDF samples were in quadrant 2 (upper left). Further details arose from the inspection of the PC loadings of the first two PCs, which highlighted the metabolites that contributed to the samples separation. Figure 9 reports loadings deriving from NMR and UV—VIS data on different scales indicated with the same color as the corresponding loadings. Block 2, with a lower number of variables, had loadings of absolute values higher than block 1. This did not mean that variables in block 2 had more importance than variables in block 1, but only that we awarded it more significance with the block-scaling pretreatment because this block consisted of a reduced number of variables and, therefore, a reduced variance. Applying block-scaling, each block has a variance of 1 and the variables of each block have the same variance.

Polyphenols, flavonoids and TAA were positively correlated with each other. Therefore, as expected, the antioxidant capacity was mainly attributed to polyphenol-type compounds. Comparing the position of samples in the score plot with the loadings position, it can be deduced that the samples derived from UF in the batch concentration process have the same metabolic composition, but different concentrations of all the metabolites were taken into account (in order of concentration: RUF > FUF > PUF). Thus, with the investigated membrane in the selected operating conditions, the ultrafiltration process allowed for preserving of the original juice composition in terms of antioxidant compounds, sugars, amino acids and organic acids. On the other hand, diafiltration changed the metabolic profile of the permeate and retentate products. The RDF samples were characterized by a higher concentration of β-carotene with respect to the other compounds, as it is a lipophilic phytochemical slightly soluble in water. Carotenoids are exciting sources in the food industry, for both their bioactive potential and colouring properties which make products more attractive to consumers [35]. When instead observing the PDF samples in the scores plot and looking at the loadings, it appears that they presented a lower content of sugars, polyphenols and flavonoids, and therefore a lower TAA, along with a higher content of amino and organic acids. At the end of each UF/DF process, the PDF and PUF permeates were mixed in the same ratio and nanofiltered. Additionally, for NF samples, the total polyphenols, flavonoids and antioxidant activity in vitro were quantified, ^1^H-NMR spectra were acquired, block-scaling pretreatment was accomplished, and PCA was performed on the fused data matrix. Figure 10 shows the biplot (score and loading plot) obtained.

It was found that 91.6% of the total variance was explained by two principal components (PCs), with PC1 accounting for 81.1% of the total variance and PC2 accounting for 10.5%. With the exception of the nanofiltration retentates, the other samples are positioned in different regions of the biplot. NF Feed samples are in the third quadrant (lower left) and present a higher concentration of sugars. Permeates deriving from the nanofiltration with NF2 membrane cluster in the fourth quadrant (lower right), having a larger amount of GABA, proline and arginine. NF1 permeates are in the first quadrant (upper right), exhibiting a higher concentration of threonine, lactic and succinic acids, aspartic acid, phlorin, isoleucine, leucine and valine. Finally, NF1 and NF2 retentates shared the second quadrant (upper left), being characterized by a close metabolite profile, mostly in terms of polyphenols, flavonoids and TAA. Therefore, under the selected operating conditions, both nanofiltration membranes were found to be suitable for the enrichment of the clarified Tacle juice in bioactive compounds.

## 4. Conclusions

High-resolution NMR spectroscopy has proven to be a powerful tool for the metabolomic and lipidomic analysis of Tacle juice, well recognized for its high content of bioactive compounds. In this work, the characterization of the juice by multinuclear (^1^H and ^13^C) 1D and 2D NMR experiments allowed us to recognize 25 metabolites in the aqueous phase and to identify triglycerides and fatty acid in the extracted organic phase. Moreover, the UV—Visible technique was applied to quantify the total content of flavonoids, polyphenols, β-carotene and TAA. In a second step of this work, membrane processes such as ultrafiltration (UF) (also in diafiltration mode) and nanofiltration (NF) were used for producing enriched fractions of bioactive compounds from the raw juice. PCA on NMR and UV—visible fused data provided graphical outputs that were easy to read and interpret, demonstrating a very effective procedure for obtaining a synthetic judgement of how UF/DF and NF membrane processes affect the metabolic profile of the juice. In addition, these techniques confirm the capability of membrane filtration processes with respect to heat treatments in preserving the chemical composition of the original juice without generating new undesirable metabolites. Further studies are underway to evaluate the performance and selectivity of UF and NF membranes in more detail.

## Figures and Tables

**Figure 1 antioxidants-12-00002-f001:**
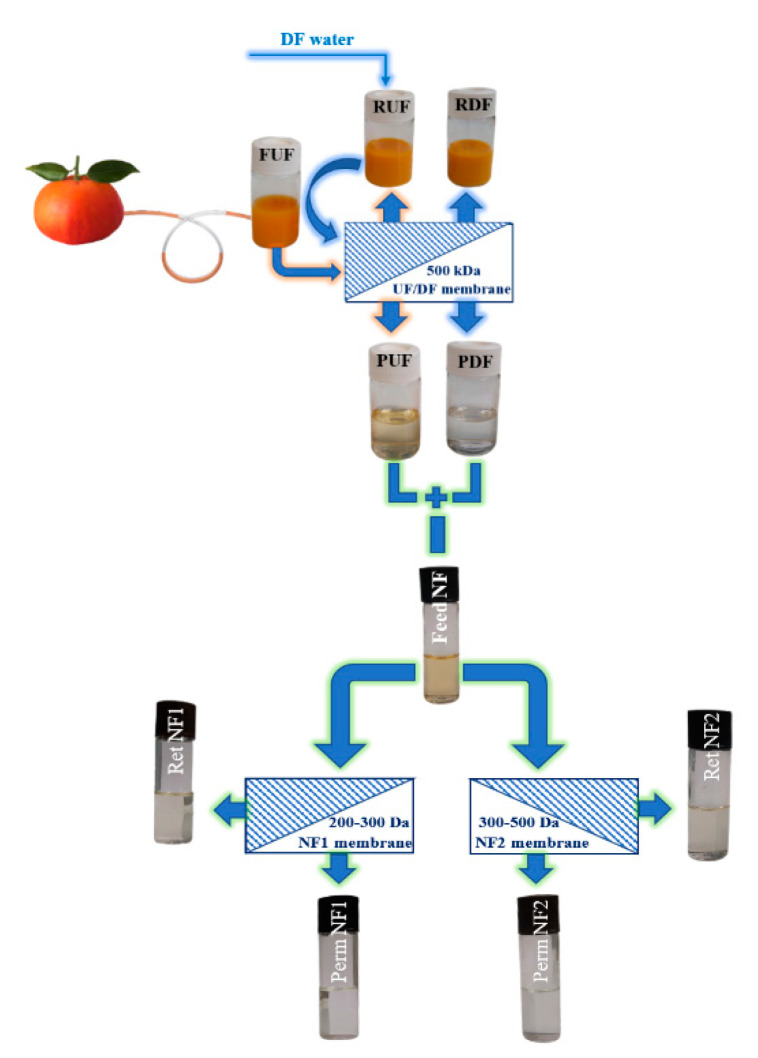
Schematic layout of the investigated process (UF, ultrafiltration; DF, diafiltration; NF, nanofiltration; F, feed; R, retentate; P, permeate).

**Figure 2 antioxidants-12-00002-f002:**
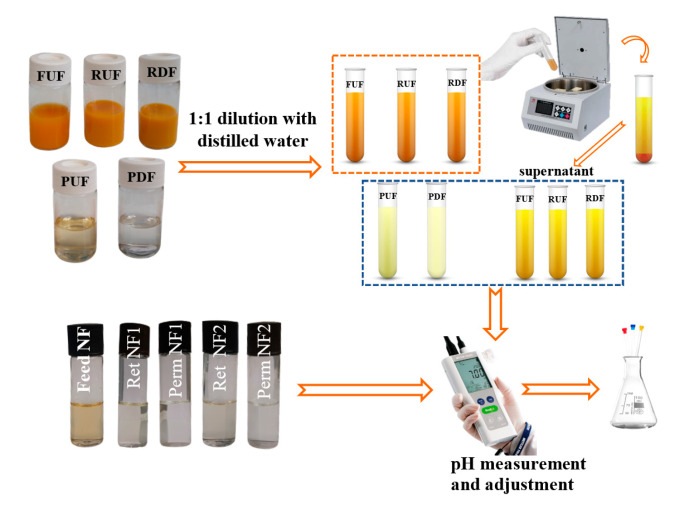
Polar phase sample preparation.

**Figure 3 antioxidants-12-00002-f003:**
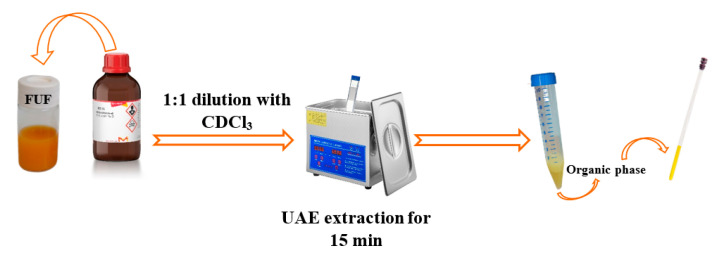
Organic phase sample preparation.

**Figure 4 antioxidants-12-00002-f004:**
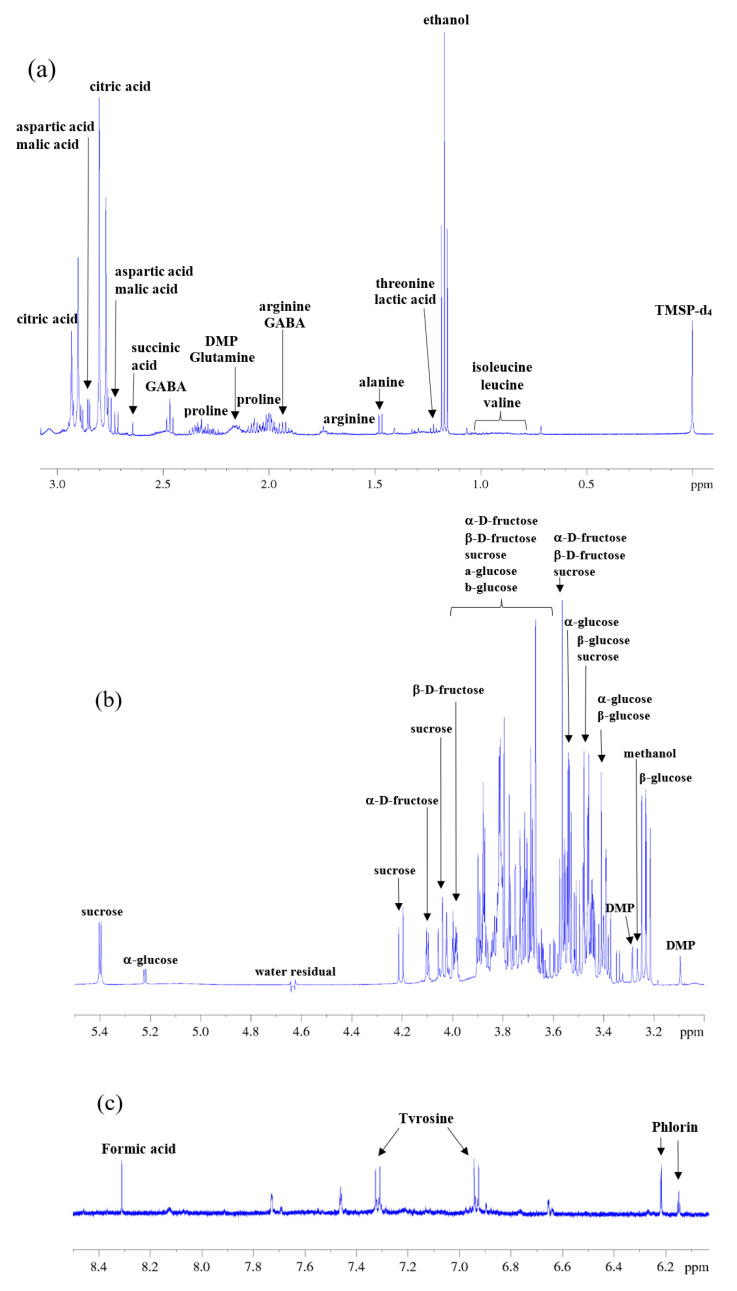
^1^H-NMR spectra obtained at 500 MHz of (**a**–**c**) the aqueous phase of Tacle juice divided into three regions (see Appendix A for assignment) and (**d**) lipophilic Tacle extract.

**Figure 5 antioxidants-12-00002-f005:**
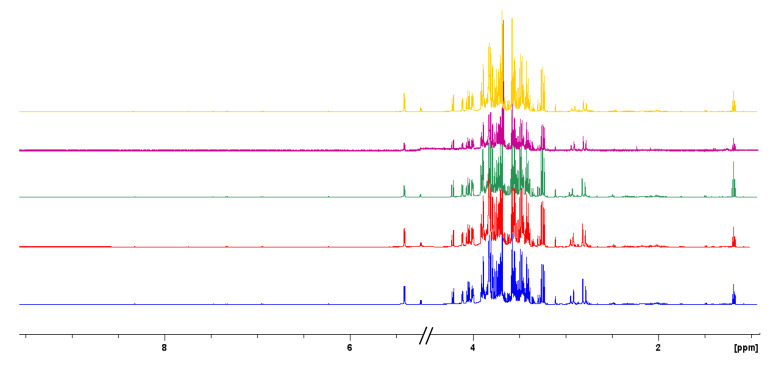
Comparison of ^1^H-NMR spectra of FUF (in blue), RUF (in red) PUF (in green), RDF (in violet), and PDF (in yellow) Tacle samples.

**Figure 6 antioxidants-12-00002-f006:**
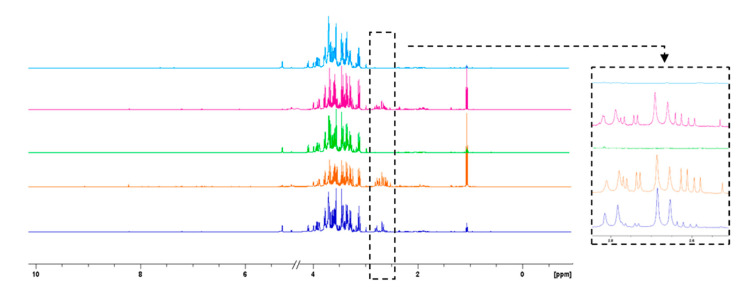
Comparison of ^1^H-NMR spectra of Feed NF (in blue), Permeate NF 1 (in orange) Retentate NF 1 (in green), Permeate NF 2 (in fuchsia) and Retentate NF 2 (in light blue) Tacle samples.

**Figure 7 antioxidants-12-00002-f007:**
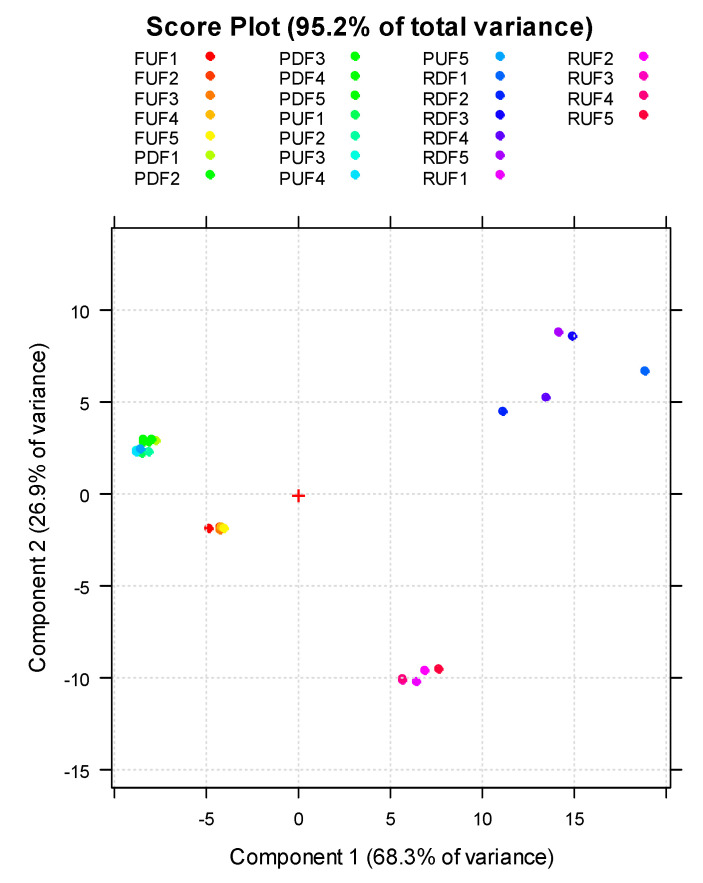
Principal component analysis score plot of the studied UF/DF samples using NMR data.

**Figure 8 antioxidants-12-00002-f008:**
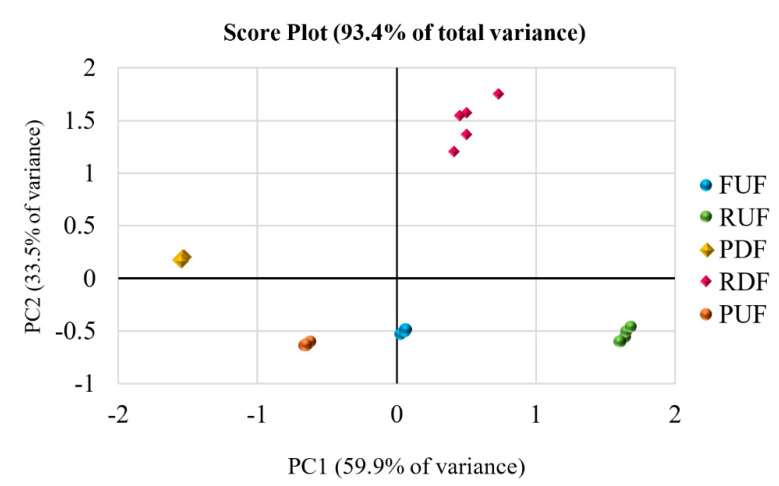
Principal component analysis score plot of the studied UF/DF samples using NMR and UV—Vis data.

**Figure 9 antioxidants-12-00002-f009:**
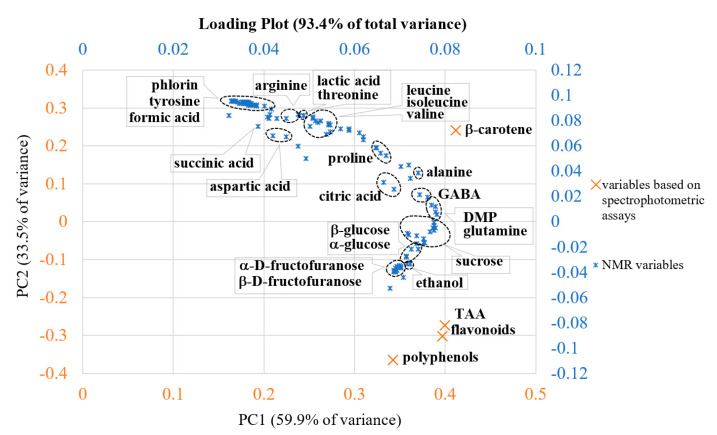
Principal component analysis loadings plot of the studied UF/DF samples using NMR and UV—Vis data.

**Figure 10 antioxidants-12-00002-f010:**
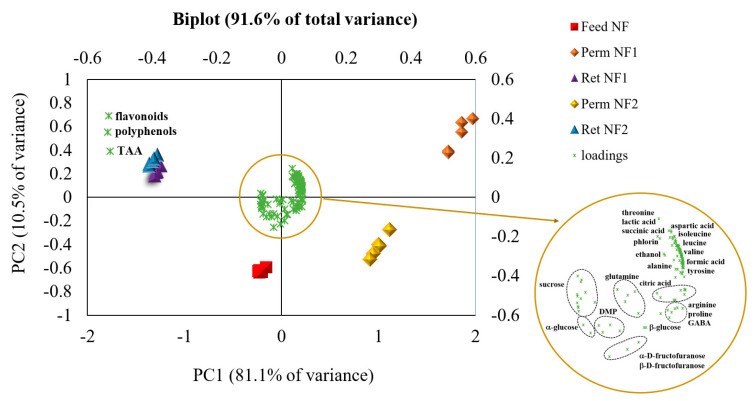
Principal component analysis biplot of the studied NF samples. The location of NMR-variables is indicated by the circle in which the assignments of the loadings are shown.

**Table 1 antioxidants-12-00002-t001:** ^1^H-NMR peak assignment of a typical lipid extract from Tacle juice. Peak labels (1 to 9) agree with those given in Figure 4d.

Peak	δ ppm	Multiplicity, *J* (Hz)	Assignment
1	0.88	t, *J* = 6.8 Hz	C*H_3_* All acyls except linolenyl
2	1.23–1.43	m	(C*H_2_*)_n_ All acyl chains
3	1.55–1.68	m	C*H_2_*CH2COOR All acyl chains
4	2.00	m	C*H_2_*CH=CH All unsaturated fatty acids
5	2.2–2.3	m	C*H_2_*COOR All acyl chains
6	3.73	q, *J* = 7.0 Hz	CH_3_C*H_2_*OH Ethanol
7	4.15 4.29	dd, *J* =12.0, 6.00 Hz dd, *J* =12.0, 6.00 Hz	C*H*_2_OCOR Glycerol (triacylglycerols)
8	5.23–5.28	m	C*H*OCOR Glycerol (triacylglycerols)
9	5.30–5.38	m	C*H*=CH All unsaturated fatty acid

## Data Availability

Data are contained within the article and Appendix A.

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
