# Peer review of "NMR-Based Characterization of Citrus Tacle Juice and Low-Level NMR and UV—Vis Data Fusion for Monitoring Its Fractions from Membrane-Based Operations"

_antioxidants, 2022, doi:10.3390/antiox12010002_

Round 1

Reviewer 1 Report

Summary: This paper describes a method for analysis of Tacle juice for its antioxidant content as well as other notable dietary metabolites. The study looks at a variety of filtration techniques and then combines NMR and UV methods of analysis to look at the variability across methods and the presence of important nutritional compounds. The authors present a clear description of their techniques and analysis. There seems to be some data points missing from the plot in Figure 10 or perhaps the description is unclear. The overall study seems appropriate and well designed, however the paper would benefit from a bit more scholarly discussion about the benefits and application of this PCA analysis within the business and inspection of tacle and other citrus juices. Some additional discussion in the conclusion about applicability and reproducibility of their findings would enhance the paper. While it may be outside the scope of this paper, since the authors mention that heat pasteurization can alter the compounds present it would help show the benefit of the filtration methods to show the reduction of nutritional compounds following heat treatment.

Language is good throughout but a few places where word choice seems wrong, worth a look over by authors and editors.

Figure 10 seems to be missing the data points for the different permeates and retentates.

It seems in looking at the figures that the groups are well defined in figure 8 so it is unclear what the gain is of applying the block scaling for figures 9 and 10.

A bit more description about how figure 9 was developed and what the reader should be getting from it would be appreciated.

Author Response

Rende, December 7th, 2022

Dear Reviewer,

Thank you very much for the effort that you spent in our manuscript. We appreciate the very constructive and helpful comments and suggestions to improve our work. In response to their input, we made the modifications recommended to the text.

We hope that our revised manuscript addresses all concerns satisfactorily.

All changes are presented in details below. The issues raised by reviewer are set in italics and our answers in plain font. All our changes are included in the revised manuscript in red color.

General comments:

  • This paper describes a method for analysis of Tacle juice for its antioxidant content as well as other notable dietary metabolites. The study looks at a variety of filtration techniques and then combines NMR and UV methods and the presence of important nutritional compounds. The authors present a clear description of their techniques and analysis. There seems to be some data points missing from the plot in Figure 10 or perhaps the description is unclear. The overall study seems appropriate and well designed, however the paper would benefit from a bit more scholarly discussion about the benefits and application of this PCA analysis within the business and inspection of tacle and other citrus juices. Some additional discussion in the conclusion about applicability and reproducibility of their findings would enhance the paper. While it may be outside the scope of this paper, since the authors mention the heat pasteurization can alter the compounds present it would help show the benefit of the filtration methods to show the reduction of nutritional compounds following heat treatment.

Authors: We would like to thank the Reviewer for this positive evaluation. By referring to Figure 10, as reported in the following, we would like to underline that in the previous version of the manuscript it was incorrectly uploaded.

A new sentence has been included in the Introduction section to highlight the benefits and application of PCA analyses within the inspection of citrus juices:

Then at line 71 of the revised version we added: PCA has also applied very successfully to the study of industrial processing of citrus fruit juices to evaluate for reliable process control, to assess the quality of orange juice and identify potential mislabeled samples.

And in the Conclusion section in relation to the unmodified composition of the original juice by using membrane filtration we added at line 426 of the revised manuscript: In addition, these techniques confirm the capability of membrane filtration processes with respect to heat treatments in preserving the chemical composition of the original juice without generating new undesirable metabolites.”

  • Language is good throughout but a few places where word choice seems wrong, worth a look over by authors and editors.

Authors:  Thank you for this suggestion.

  • Figure 10 seems to be missing the data points for the different permeates and retentates.

Authors: We noted that in the original submission Figure 10 was not correctly uploaded. Indeed, in this figure data points for all samples were missing. We have included the correct version of the Figure in the revised version of the manuscript. In addition, we noted that in the original version the fractions obtained using the nanofiltration membrane NF1 were wrongly indicated as “Perm TS40” and “Ret TS40”, while those obtained using the nanofiltration membrane NF2 were reported as “Perm XN45” and “Ret XN45”. The legend of Figure 10 has been corrected in the revised version of the manuscript.

The correct version of Figure 10 is reported in the following and replace the old figure at line 398.

Figure 10. Principal component analysis biplot of the studied NF samples. The location of NMR-variables is indicated by the circle of which the zoom and the assignments of the loadings are shown.

  • It seems in looking at the figures that the groups are well defined in figure 8 so it is unclear what the gain is of applying the block scaling for figures 9 and 10.

Authors: Concerning this comment, we would like to clarify some points to make this answer clearer. As reported in the manuscript, block-scaling is used when dealing with two (or more) block of variables, in which a large block of variables dominates over a smaller block of variables for purely numeric reason. In this case, the larger block of variables is represented by NMR data, while the smaller block is constituted by UV-Vis data. The use of autoscaling pretreatment on our fused data matrices, would be not suitable because the quantitative information of the measured UV-Vis variables would be masked by variation in the other NMR variables. Thus, to address this problem we employed the block-scaling as pretreatment to achieve a low-level NMR and UV-Vis data fusion. Figure 8 and 9 are both outputs of PCA statistical analysis performed with a previous block-scaling of UF samples’ data. Figure 8 shows the score plot, thus define the location of our samples in the new principal components, while Figure 9 shows the loading plot in which we can underline the location of our initial variables in the new principal components. The joint analysis of figure 8 and figure 9 allows us to understand which are the variables that determine the differentiation and separation of the samples into defined groups.  On the other hand, figure 10 shows the biplot (score plot + loading plot) obtained by performing block-scaling and PCA on NF samples. Therefore, in summary, all the figures (8, 9 and 10) have been obtained with the block-scaling pretreatment, but while figures 8 and 9 refer to the ultrafiltration process, figure 10 refers to the nanofiltration processes. Therefore, in the manuscript, we want to emphasize that block-scaling is necessary to process more data blocks. We also want to underline that using more than one analytical technique can be essential to distinguish the samples better if the two techniques provide complementary information.

  • A bit more description about how figure 9 was developed and what the reader should be getting from it would be appreciate.

Authors: Figure 9 is the PCA loadings plot, thus one of the two outputs obtained by performing PCA of UF samples. The multivariate statistical analyse was performed by the Chemometric Agile Tool (CAT) as reported in the Materials and methods section (lines 237-238 of the revised version).

The information that the reader could get from figure 9, has been already reported in lines 348-349 of the revised version: “Further details arose from the inspection of the PC loadings of the first two PCs which highlighted the metabolites that contributed to the samples separation.”   

Moreover, we already explain the information that reader should get from loadings (Figure 9) combined to scores (Figure 8) as can be read at line 361 to line 376 of the revised version:

“Polyphenols, flavonoids and TAA were positively correlated among them. Therefore, as expected, the antioxidant capacity was mainly attributed to polyphenol-type compounds. Comparing the position of samples in the score plot with the loadings position it can be figure out that the samples deriving from UF in batch concentration process have the same metabolic composition, but different concentration of all the metabolites took into account (in order of concentration: RUF>FUF>PUF). Thus, with the investigated membrane in the selected operating conditions, the ultrafiltration process allowed for preserving the original juice composition in terms of antioxidant compounds, sugars, amino acids and organic acids. On the other hand, diafiltration changed the metabolic profile of the permeate and retentate products. The RDF samples were characterized by a higher concentration of b-carotene with respect to the other compounds, being a lipophilic phytochemical slightly soluble in water. Carotenoids are exciting sources in the food field, both for their bioactive potential and colouring properties, making products more attractive to consumers [35]. Observing, instead, the PDF samples in the scores plot and looking at the loadings it emerges that they presented a lower content of sugars, polyphenols, flavonoids, and therefore lower TAA, and higher content of amino and organic acids.”

Reviewer 2 Report

This paper proposes a rationalization based on NMR analyses of the different filtration types of natural juices containing antioxidants. It  is well written and the reading is easy. 

Moreover, some information could be useful for the reader

- What is the delay between juice production and the experiment. Indeed, the antioxidant could be unstable.

- What is the yield of each filtration step compared to feed juice ?

Author Response

Rende, December 7th, 2022

Dear Reviewer,

Thank you very much for the effort that you spent in our manuscript. We appreciate the very constructive and helpful comments and suggestions to improve our work. In response to their input, we made the modifications recommended to the text.

We hope that our revised manuscript addresses all concerns satisfactorily.

All changes are presented in details below. The issues raised by reviewer are set in italics and our answers in plain font. All our changes are included in the revised manuscript in red color.

General comments:

This paper proposes a rationalization based on NMR analyses of the different filtration types of natural juices containing antioxidants. It is well written and the reading is easy.

Authors: We would like to thank the Reviewer for this positive evaluation.

Moreover, some information could be useful for the reader

  • What is the delay between juice production and the experiment. Indeed, the antioxidant could be unstable.
  • What is the yield of each filtration step compared to feed juice?

Authors: We would like to thank the Reviewer for shedding light on these very important points thus giving us the opportunity to improve the work by adding these informations. Therefore, following the Reviewers’ suggestions, we have added in the subsection 2.1 Tacle juice and juice processing, at line 124 of revised version, the following sentence to satisfy suggestion 1):

Each batch, previously frozen at a temperature of -18° C, was thawed after about two weeks to be subjected to membrane processes and the resulting fractions were then frozen again to be thawed in view of the analysis of chemical characteristics: all samples collected from juice processing, for a total of 50 samples, were spectroscopically and spectrophotometrically analyzed and statistically processed by PCA.”

And, in the same subsection at line 106 and at line 121, the following sentences have been included to fulfil suggestion 2):

“The UF process was operated in selected operating conditions according to the batch concentration configuration in order to clarify the juice up to a volume reduction factor (VRF, defined as the ratio between the initial feed volume and the volume of the resulting retentate) of 4.08.”

“The NF processes were operated up to a VRF of 3.5.”
